# Hydrogen-bonded organic framework@conductive metal-organic framework heterostructures for ampere-level hydrogen peroxide production

Yingying Zou[1], Yulin Zhang[1], Chaoqi Zhang[1], Tong Bao[1], Yamin Xi[1], Niqi Ao[1], Zhijie Li[1], Yunying Wang[1], Chao Liu [1,2,3] ✉ & Chengzhong Yu [1,2,4] ✉

Electrochemical two-electron oxygen reduction reaction (2e$^-$ ORR) in neutral environments holds remarkable promise for sustainable hydrogen peroxide ($H_2O_2$) production. However, its practical application is largely hindered due to the scarcity of electrocatalysts with high selectivity and durability under ampere-level current densities. Herein, a hydrogen-bonded organic framework@conductive metal-organic framework (HOF@cMOF) heterostructure is designed for industrial-level $H_2O_2$ electrosynthesis. Through integrating DAT-HOF (DAT=diaminotriazole) and Co-cMOF, Co-N bonds formed at the heterointerface modulates the electronic structure of Co sites, optimizing the adsorption strength of oxygen intermediates with improved activity and selectivity. Besides, the formation of built-in electric field drives the proton migration from DAT-HOF to Co-cMOF, facilitating the $O_2$ protonation to $H_2O_2$ at Co sites. In further combination with the high proton donation capability of DAT-HOF and high conductivity of Co-cMOF, efficient $H_2O_2$ production is achieved with a $H_2O_2$ Faradic efficiency of $97.1 \pm 0.4\%$, a $H_2O_2$ yield of 738.9 mg h$^{-1}$ cm$^{-2}$ and a long-term durability over 100 h at 1200 mA cm$^{-2}$. This work offers a high-performance electrocatalyst for promoting the industrial implementation of $H_2O_2$ electrosynthesis.

Hydrogen peroxide ($H_2O_2$) is a vital chemical extensively used in pharmaceuticals, paper manufacturing, chemical synthesis and environmental protection[1,2]. As a promising alternative to the traditional anthraquinone method, the electrochemical two-electron oxygen reduction reaction (2e$^-$ ORR) process offers a sustainable and decentralized approach for $H_2O_2$ production across a wide pH range from 0–14[3–5]. However, the widely investigated reaction systems in acidic and alkaline mediums have limitations, including the $H_2O_2$ instability, corrosiveness and potential environmental issues[6–8]. In contrast, 2e$^-$ ORR operated in mild neutral condition features environmental friendliness, low corrosion, high stability and flexible utilization of produced $H_2O_2$[9–11]. From a practical perspective, achieving neutral-condition $H_2O_2$ production via 2e$^-$ ORR at ampere-level current densities (>300 mA cm$^{-2}$)[12–14] is of great significance. To date, various electrocatalysts such as noble metals[15], carbons[16], transition metal compounds[17], single-atom materials[18,19] and metal-organic frameworks (MOFs)[20–22] have been applied for this purpose. However, most of them demonstrate high Faradaic efficiency (FE) only at modest current

[1]School of Chemistry and Molecular Engineering, East China Normal University, Shanghai, China. [2]State Key Laboratory of Petroleum Molecular and Process Engineering, SKLPMPE, East China Normal University, Shanghai, China. [3]Shanghai Frontiers Science Center of Molecule Intelligent Syntheses, School of Chemistry and Molecular Engineering, East China Normal University, Shanghai, China. [4]Australian Institute for Bioengineering and Nanotechnology, The University of Queensland, Brisbane, Queensland, Australia. ✉e-mail: cliu@chem.ecnu.edu.cn; c.yu@uq.edu.au

densities (<100 mA cm$^{-2}$). The development of electrocatalysts that are capable of efficiently driving 2e$^-$ ORR at high current densities up to ampere levels remains a challenge.

Among currently studied 2e$^-$ ORR electrocatalysts, MOFs with large specific surface area, high porosity and tunable structures have garnered particular interest[23,24]. Compared to conventional MOFs, conductive MOFs (cMOFs) with π-conjugated frameworks exhibit high electrical conductivity and are ideal candidates for the design of advanced 2e$^-$ ORR electrocatalysts[25–27]. However, the reported cMOF-based 2e$^-$ ORR electrocatalysts (e.g., Mg-HITP[28], ZnCu-MOF (H)[29], Ni-HAB[30]) suffer from unsatisfactory performances in neutral electrolytes even at relatively low current densities, thus their application at industrial-relevant ampere-level current densities is rarely reported. The 2e$^-$ ORR pathway involves a two-proton coupled two-electron transfer process ($O_2 + H^+ + e^- \rightarrow {}^*OOH$, ${}^*OOH + H^+ + e^- \rightarrow H_2O_2$) and competes with the 4e$^-$ ORR pathway to generate $H_2O$[31,32]. The precise modulation of catalytic active sites to suppress the 4e$^-$ ORR pathway is essential for the selective production of $H_2O_2$. In addition to improving the selectivity of 2e$^-$ ORR against 4e$^-$ ORR, the proton supply is a critical factor that determines the formation of $H_2O_2$ in neutral environments[33,34]. Insufficient proton supply is detrimental to the generation of ${}^*OOH$ and the conversion of ${}^*OOH$ to $H_2O_2$, reducing the 2e$^-$ ORR activity and selectivity[35]. At ampere-level current densities, the faster proton consumption imposes even higher demand on the proton supply[33]. To achieve ampere-level $H_2O_2$ electrosynthesis, the rational design of cMOF-based electrocatalysts with both modulated active sites and sufficient proton supply capability is an essential but challenging task.

As another important class of porous crystalline materials, hydrogen-bonded organic frameworks (HOFs) are formed by intermolecular hydrogen bonding (H-bond) of organic building blocks[36–38]. The versatile surface functional groups of HOFs enable effective hybridization with other materials, yielding heterostructures with synergistic properties[39]. Moreover, the chemical interaction at the heterointerface can induce charge redistribution and modulate the electronic structure of active sites, thereby enhancing the intrinsic activity and selectivity[40]. Importantly, the abundant hydrogen-bond networks in HOFs can facilitate proton conduction, potentially

enhancing local proton donation during electrocatalysis[41]. Thus, we hypothesize that the construction of cMOFs/HOFs heterostructures is a promising approach to develop high-performance 2e$^-$ ORR electrocatalysts. Such a design has been rarely reported.

Herein, a HOF@cMOF heterostructure is constructed for ampere-level $H_2O_2$ electrosynthesis, exhibiting a $H_2O_2$ FE of 97.1 ± 0.4% and a $H_2O_2$ production rate of 738.9 mg h$^{-1}$ cm$^{-2}$ at 1200 mA cm$^{-2}$, along with a long-term stability over 100 h. The conductive Co-HHTP (HHTP = 2,3,6,7,10,11-hexahydroxytripehenylene) is grown on DAT-HOF (DAT=diaminotriazole) with sufficient proton donation ability, resulting in a unique rod-on-rod heterostructure (Fig. 1a). At the HOF-cMOF heterointerface, the formation of Co-N bonds results in the electron-enriched Co active sites with optimized adsorption strength of oxygen intermediates, making pivotal contributions to the enhanced 2e$^-$ ORR activity and selectivity (Fig. 1b). Furthermore, the charge redistribution between two components with different Fermi levels induces the formation of a built-in electric field, driving the directional migration of protons from DAT-HOF to Co-HHTP for facilitating the protonation of $O_2$ to $H_2O_2$ at Co sites (Fig. 1c). By integrating modulated active sites, high electron conductivity and enhanced hydrogenation capability, the rationally designed heterostructure delivers excellent 2e$^-$ ORR performance at ampere-level current densities, superior than most reported electrocatalysts. Our work paves the way for the design of advanced electrocatalysts for practical $H_2O_2$ production.

## Results

To prepare the DAT-HOF@Co-HHTP heterostructure, DAT-HOF was firstly synthesized through the imidization reaction according to a reported protocol[42,43]. Scanning electron microscope (SEM) and transmission electron microscopy (TEM) images of DAT-HOF (Supplementary Fig. 1a, b) show a uniform one-dimensional (1D) rod-like morphology with micrometer-scale length and smooth surface. The crystalline structure of DAT-HOF was studied by powder X-ray diffraction (PXRD). The experimental PXRD pattern (Supplementary Fig. 1c) matches well with the simulated profile via Pawley refinement ($R_p$ = 5.26% and $R_{wp}$ = 6.76%), revealing a hexagonal crystal structure with cell parameters of $a$ = 14.25, $b$ = 14.25 Å, $c$ = 5.17 Å.

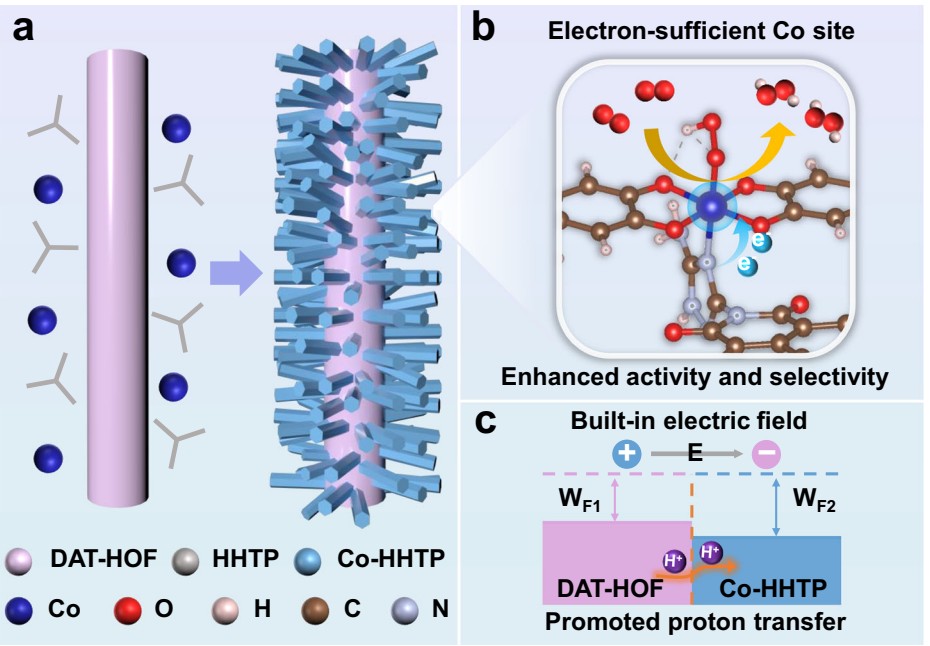

**Fig. 1 | Synthesis and working mechanism of DAT-HOF@Co-HHTP.** Schematic illustration of the **a** synthesis process, **b** interfacial active site and **c** built-in electric field of DAT-HOF@Co-HHTP heterostructure.

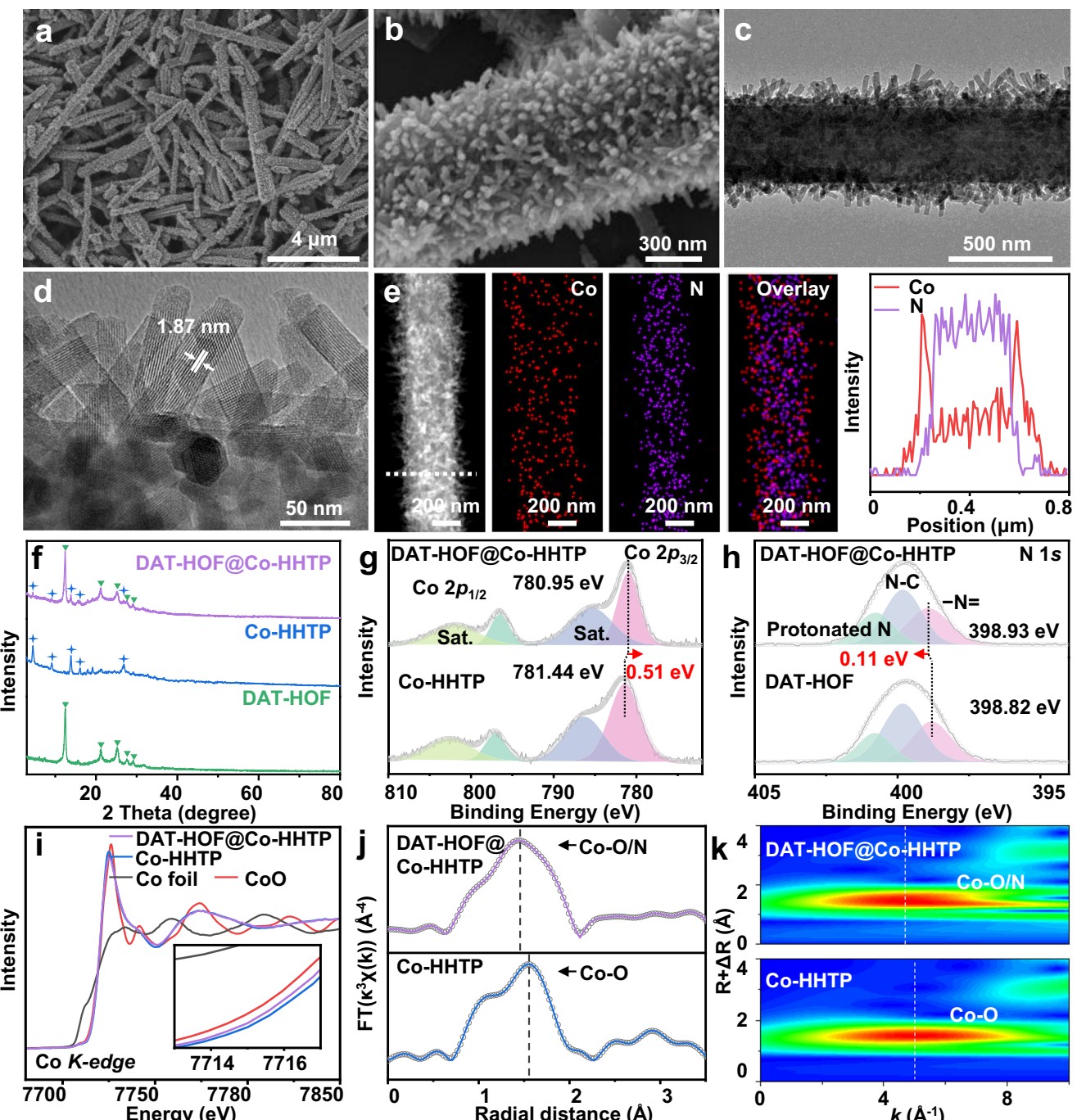

**Fig. 2 | Morphology and structural characterization. a**, **b** SEM, **c** TEM, **d** HRTEM, and **e** HADDF-STEM and corresponding EDS mapping images of DAT-HOF@Co-HHTP. Inset in **e** is the line scanning spectra. **f** XRD patterns of DAT-HOF@Co-HHTP, DAT-HOF and Co-HHTP. **g** Co 2*p* spectra of DAT-HOF@Co-HHTP and Co-HHTP and **h** N 1*s* spectra of DAT-HOF@Co-HHTP and DAT-HOF. **i** Co K-edge of XANES spectra, **j** Co K-edge FT-EXAFS spectra, and **k** wavelet transforms for Co K-edge EXAFS of DAT-HOF@Co-HHTP and Co-HHTP. Source data for Fig. 2 are provided as a Source Data file.

The subsequent reaction of DAT-HOF with cobalt (II) acetate and HHTP ligands resulted in the DAT-HOF@Co-HHTP heterostructure. The low-magnification SEM image of DAT-HOF@Co-HHTP (Fig. 2a) shows a well-preserved microrod-like morphology with rough surface. At higher magnification (Fig. 2b), high-density nanorods with an average diameter of ~30 nm as the shell adhered to the microrod as the core are observed. The core-shell structured rod-on-rod heterostructure is further demonstrated by TEM studies (Fig. 2c). In the high-resolution TEM (HRTEM) image of the shell nanorods (Fig. 2d), the distinct lattice fringes with an interplanar distance of 1.87 nm are ascribed to the (100) planes of Co-HHTP. Figure 2e displays the high-

angle annular dark-field scanning TEM (HAADF-STEM) and energy-dispersive X-ray spectroscopy (EDX) element mapping images of DAT-HOF@Co-HHTP. The Co and N elements predominantly exist in the internal core and external shell regions, respectively. The line scanning spectra collected along the short axis (inset in Fig. 2e) further indicate the N-rich DAT-HOF core and Co-rich Co-HHTP shell in the rod-on-rod heterostructure.

To analyze the crystalline structure and chemical composition of DAT-HOF@Co-HHTP, XRD and Fourier transform infrared (FTIR) spectroscopy measurements were conducted. In addition to DAT-HOF, Co-HHTP nanorods were also prepared as a control

(Supplementary Fig. 2). As shown in Fig. 2f, two groups of diffraction peaks attributed to DAT-HOF and Co-HHTP are simultaneously observed in the XRD pattern of DAT-HOF@Co-HHTP. In the FTIR spectrum of DAT-HOF (Supplementary Fig. 3a), the peaks located at 1410 and 1060 $cm^{-1}$ are attributed to the axial C=N−C stretching vibration of imine and in-plane rocking vibration of −$NH_2$[43]. Besides, the peak at 1657 $cm^{-1}$ corresponds to the stretching mode of C=$NH^+$ bond, originated from the protonation of C=N group in DAT-HOF (Supplementary Fig. 3b)[44]. Compared to DAT and NTD ligands, the additional peaks at 3247 and 3066 $cm^{-1}$ in the spectrum of DAT-HOF are associated with the stretching vibration of H-bond N−H (Supplementary Fig. 4), indicating the formation of hydrogen bonding in DAT-HOF[45,46]. For Co-HHTP, the peaks at 1454, 1298 and 1216 $cm^{-1}$ in the spectrum are assigned to the benzene skeleton vibrations of HHTP[47]. By integrating DAT-HOF and Co-HHTP, the typical peaks of two components are simultaneously detected in DAT-HOF@Co-HHTP. Supplementary Fig. 5 shows the $N_2$ adsorption-desorption isotherms of the samples. The Brunauer-Emmett-Teller (BET) specific surface areas and pore volumes of DAT-HOF, Co-HHTP and DAT-HOF@Co-HHTP were determined to be 75.1 $m^2 g^{-1}$ and 0.29 $cm^3 g^{-1}$ for DAT-HOF, 91.9 $m^2 g^{-1}$ and 0.46 $cm^3 g^{-1}$ for DAT-HOF@Co-HHTP, and 118.3 $m^2 g^{-1}$ and 0.50 $cm^3 g^{-1}$ for Co-HHTP, respectively.

X-ray photoelectron spectroscopy (XPS) study was employed to investigate the surface chemical states of the three samples. The survey spectra (Supplementary Fig. 6) show the co-existence of Co, O, N and C elements in DAT-HOF@Co-HHTP heterostructure. The Co $2p$ spectrum of Co-HHTP (Fig. 2g) was deconvoluted into Co $2p_{1/2}$ and Co $2p_{3/2}$ orbitals of $Co^{2+}$ at 797.04 and 781.44 eV, respectively, with two satellite peaks at 802.62 and 786.38 eV. Compared to Co-HHTP, the binding energy of Co $2p_{3/2}$ orbital of $Co^{2+}$ in DAT-HOF@Co-HHTP decreases by 0.51 eV. In the N $1s$ spectrum of DAT-HOF (Fig. 2h), three main peaks are observed at 398.82, 399.81 and 400.80 eV, corresponding to −N=, N−C and protonated N[44], respectively, consistent with the FTIR observations. After hybridization with Co-HHTP, the peak positions of N−C and protonated N are almost unchanged, while the binding energy of −N= peak is elevated by ≈0.11 eV. The opposite change trend of N $1s$ and Co $2p$ indicates the electron transfer from DAT-HOF to Co-HHTP via the Co−N bonds that are formed by the interaction between $Co^{2+}$ and −N=.

X-ray absorption spectroscopy (XAS) measurements were carried out to further elucidate the electronic structure of DAT-HOF@Co-HHTP, with Co-HHTP as comparison. As shown in the X-ray absorption near-edge structure (XANES) spectra (Fig. 2i), the energy position of Co K-edge for DAT-HOF@Co-HHTP is more negative than Co-HHTP, indicating the formation of electron-enriched Co sites in the heterostructure via electron transfer from DAT-HOF to Co-HHTP. Figure 2j presents the Fourier transformed-extended X-ray absorption fine structure (FT-EXAFS) spectra of DAT-HOF@Co-HHTP and Co-HHTP. In the Co K-edge spectrum of Co-HHTP, the main peak at ≈1.58 Å is assigned to the Co−O bond, which originates from the coordination of $Co^{2+}$ with HHTP ligands and $H_2O$. For DAT-HOF@Co-HHTP, the peak exhibits a slight negative shift of 0.32 Å compared to Co-HHTP, possibly resulting from the formation of Co−N bond at DAT-HOF/Co-HHTP interface. The hypothesis is further supported by the wavelet transform profiles derived from the EXAFS spectra (Fig. 2k). The XAS observations are well consistent with XPS results, collaboratively verifying the successful construction of chemically bonded DAT-HOF@Co-HHTP heterostructure.

The ORR catalytic performance of DAT-HOF@Co-HHTP heterostructure for $H_2O_2$ production was evaluated using a three-electrode cell in an $O_2$-saturated 0.1 M phosphate buffer saline (PBS) solution (pH = 7). Linear sweep voltammetry (LSV) measurement was first performed using a rotating ring-disk electrode (RRDE) system at 1600 rpm. The ORR currents were recorded at the disk electrode (solid line) and the oxidation of generated $H_2O_2$ was detected via the Pt ring electrode (dashed line). A shown in the LSV curves (Fig. 3a), DAT-HOF@Co-HHTP exhibits the most positive onset potential (the potential at 0.10 mA $cm^{-2}$) of 0.67 V vs. RHE among all the samples, indicative of the highest ORR activity. Besides, the Tafel slope of DAT-HOF@Co-HHTP was determined to be 71.3 mV $dec^{-1}$, lower than that of Co-HHTP (89.4 mV $dec^{-1}$) and DAT-HOF (132.8 mV $dec^{-1}$, Supplementary Fig. 7a), suggesting the fastest reaction kinetics of DAT-HOF@Co-HHTP. The $2e^-$ ORR selectivity was calculated based on the disk and ring currents, then plotted as a function of the applied potential in Fig. 3b. The $2e^-$ ORR selectivity of DAT-HOF@Co-HHTP exceeds 90.0% across a wide potential range from 0.0 to 0.6 V vs. RHE with a peak value of 99.3% at 0.6 V vs. RHE, superior than Co-HHTP (54.4% at 0.6 V vs. RHE) and DAT-HOF (36.4% at 0.6 V vs. RHE). The electron transfer numbers (n) of DAT-HOF, Co-HHTP and DAT-HOF@Co-HHTP were calculated to be 3.32, 2.89 and 2.08, respectively (Supplementary Fig. 7b), manifesting the high $2e^-$ ORR selectivity of DAT-HOF@Co-HHTP.

Moreover, the LSV polarization curves measured in the flow electrolytic cell system (Supplementary Figs. 8 and 9) show that DAT-HOF@Co-HHTP exhibits increased ORR current density and activity than DAT-HOF and Co-HHTP, consistent with the results obtained from the RRDE measurements. The $H_2O_2$ production rate and Faraday efficiency (FE) were further calculated as displayed in Fig. 3c, d, respectively. DAT-HOF@Co-HHTP exhibits increased $H_2O_2$ production rates from 125.3 mg $h^{-1} cm^{-2}$ at 200 mA $cm^{-2}$ to 738.9 mg $h^{-1} cm^{-2}$ at 1200 mA $cm^{-2}$, all with a high $H_2O_2$ FE (97.1 ± 0.4% to 99.0 ± 0.6%). At a higher current density of 1400 mA $cm^{-2}$, even with a slight drop of 5.9% for $H_2O_2$ FE (still exceeding 90%), a $H_2O_2$ generation rate of 809.6 mg $h^{-1} cm^{-2}$ is achieved, suggesting the efficient conversion of $O_2$ to $H_2O_2$ at ampere level. For Co-HHTP, the $H_2O_2$ production rate is elevated at relatively low current densities from 200 to 600 mA $cm^{-2}$, but quickly reaches a plateau after 600 mA $cm^{-2}$ with a peak value of 134.9 mg $h^{-1} cm^{-2}$. In contrast to the well-maintained and high $H_2O_2$ FE values of DAT-HOF@Co-HHTP, the $H_2O_2$ FE of Co-HHTP is dramatically decreased from 39.6 ± 0.4% to 15.2 ± 0.1%. For DAT-HOF, the overall $H_2O_2$ production rate remains limited (-1.7 to 7.7 mg $h^{-1} cm^{-2}$).

Notably, the performance of DAT-HOF@Co-HHTP especially at ampere level is superior to most reported $2e^-$ ORR electrocatalysts including noble metal nanoparticles, transition metal compounds, metal-free carbon materials and other MOFs (Fig. 3e and Supplementary Table 1). Moreover, even after 100 h of continuous electrolysis, the $H_2O_2$ FE and production rate of DAT-HOF@Co-HHTP decreased by only 1.6% and 11.6 mg $h^{-1} cm^{-2}$, respectively (Fig. 3f). Additionally, the chronoamperometry (I-t) curve shows a nearly constant current density during a long-period operation of 100 h (Supplementary Fig. 10), further demonstrating the good electrochemical stability of DAT-HOF@Co-HHTP. Additionally, DAT-HOF@Co-HHTP after test was collected and characterized by SEM, TEM, XRD and XPS (Supplementary Fig. 11). The results show that the rod-on-rod morphology, crystal and electronic structure are well preserved, demonstrating the long-term durability.

By further adding $Na_2CO_3$ into the electrolyte after a successive electrolysis of 100 h, the produced $H_2O_2$ can be extracted in the form of sodium carbonate perhydrate ($Na_2CO_3 \cdot 1.5H_2O_2$) as a solid $H_2O_2$ product. The XRD pattern of collected powder shows well-matched diffraction peaks with those of standard pattern, verifying the formation of solid $H_2O_2$ (Fig. 3g)[48]. The variation in relative peak intensity may be attributed to the difference in preferred orientation between solid and standard sample. Additionally, the Raman spectra of solid $H_2O_2$ and standard $Na_2CO_3 \cdot 1.5H_2O_2$ (Supplementary Fig. 12) show almost identical patterns, further verifying the successful preparation of solid $H_2O_2$. When directly used as an oxidant, the as-synthesized solid $H_2O_2$ powder exhibits high oxidative activity for efficient degradation of various organic dyes (Fig. 3g). The preparation of solid $H_2O_2$ offers a facile approach for reducing the storage and transportation

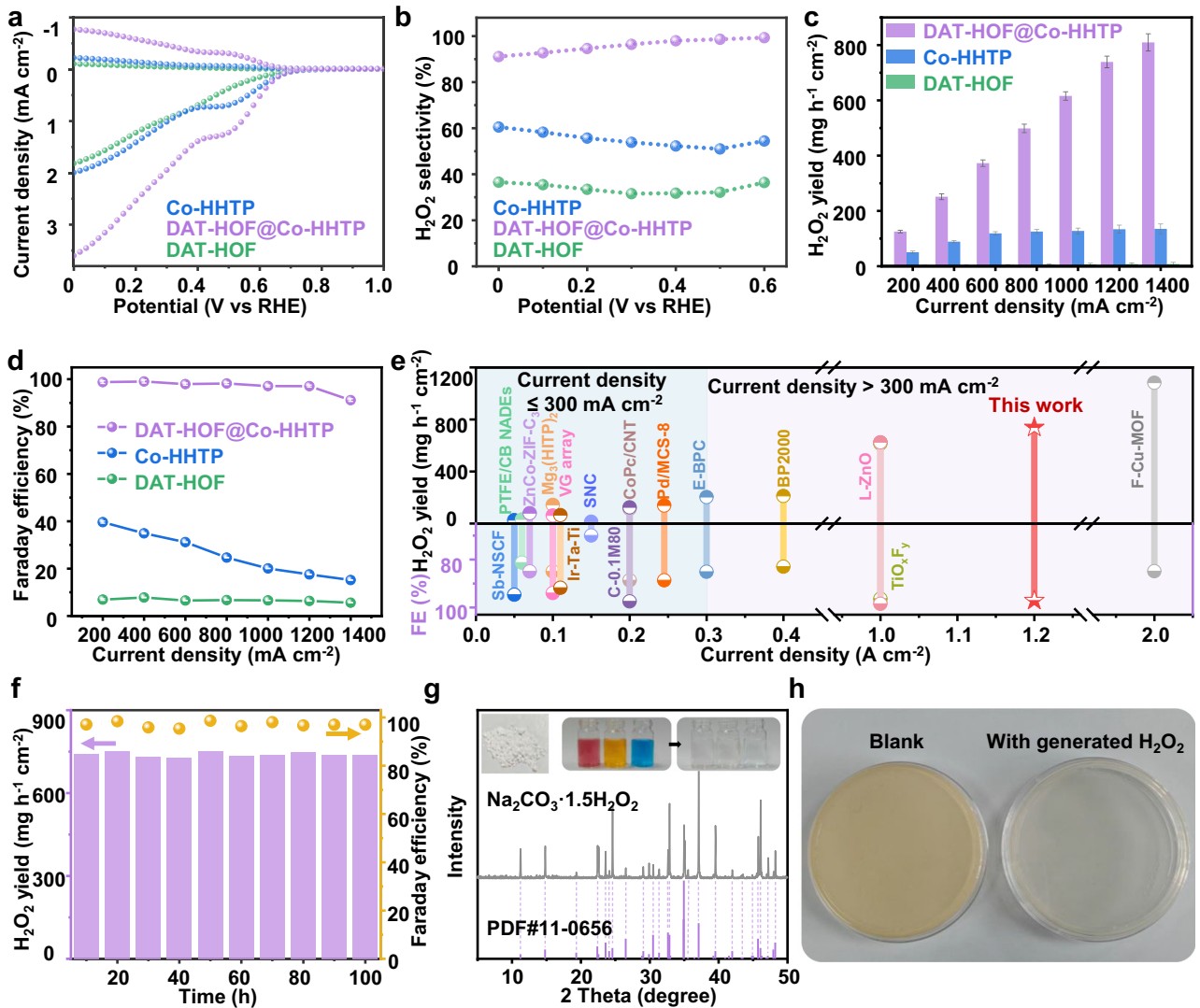

**Fig. 3 | Electrocatalytic 2e⁻ ORR performance. a** LSV polarization curves, **b** $H_2O_2$ selectivity, **c** $H_2O_2$ production rate and **d** Faraday efficiency of DAT-HOF@Co-HHTP, DAT-HOF and Co-HHTP. **e** Performance comparison with reported electrocatalysts. **f** $H_2O_2$ FE and production rate versus time at a current density of 1200 mA cm⁻². **g** XRD pattern of extracted solid $H_2O_2$ (insets are photographs of solid $H_2O_2$ and degradation of different dyes by solid $H_2O_2$). **h** In situ generated $H_2O_2$ for disinfection of Staphylococcus aureus. Error bars represent the standard error of the mean from three independent experimental measurements. All the curves were used without IR compensation. Source data for Fig. 3 are provided as a Source Data file.

costs of liquid $H_2O_2$[49]. Moreover, the generated $H_2O_2$ can inhibit the growth of Staphylococcus aureus, indicating its superior antibacterial application potential (Fig. 3h).

To further evaluate the electrochemical property, the measurements of electrochemical surface area (ECSA) and electrochemical impedance spectroscopy (EIS) were conducted. As determined by the double-layer capacitances ($C_{dl}$) in the non-Faradic region (1.03–1.13 V vs. RHE, Supplementary Fig. 13), the ECSA values of the samples show the trend of DAT-HOF@Co-HHTP (0.30 mF cm⁻²) > Co-HHTP (0.18 mF cm⁻²) > DAT-HOF (0.08 mF cm⁻²), indicating the higher intrinsic activity of DAT-HOF@Co-HHTP[20]. The EIS spectra (Supplementary Fig. 14) show that the charge-transfer resistance ($R_{ct}$) of DAT-HOF@Co-HHTP (25.4 Ω cm²) is slightly larger than that of Co-HHTP (20.6 Ω cm²), but significantly lower than DAT-HOF (59.7 Ω cm²), suggesting the enhanced charge transportation capability of DAT-HOF@Co-HHTP than DAT-HOF by the integration with conductive Co-HHTP.

In situ attenuated total reflection infrared spectroscopy (ATR-IR) was applied to monitor the formation of adsorbed oxygen intermediates on DAT-HOF@Co-HHTP and Co-HHTP during the ORR

process. As shown in the ATR-IR spectrum of DAT-HOF@Co-HHTP at open circuit potential (OCP, Fig. 4a and Supplementary Fig. 15a), the peak at 1488 cm⁻¹ is attributed to the O−O stretching mode of adsorbed molecular oxygen (*$O_2$)[50]. With the applied potential decreased to 0.5 V vs. RHE, the peak intensity of *$O_2$ diminishes with the generation of two new peaks of *OOH and *HOOH at 1255 and 1396 cm⁻¹, respectively, indicating the conversion of $O_2$ to oxygen intermediates[50]. By further lowering the potential from 0.5 to 0.1 V vs. RHE, *$O_2$ is dramatically consumed with the accumulation of *OOH and *HOOH. Compared to DAT-HOF@Co-HHTP, the peak intensity of *$O_2$ at OCP for Co-HHTP is obviously weaker, revealing the enhanced $O_2$ adsorption on DAT-HOF@Co-HHTP (Fig. 4b and Supplementary Fig. 15b). At more negative potentials, the conversion of *$O_2$ to *OOH and *HOOH is significantly retarded, suggesting the promoted 2e⁻ ORR process over DAT-HOF@Co-HHTP, in accordance with the electrocatalytic results.

To understand the enhanced performance of DAT-HOF@Co-HHTP, the proton supply and transfer properties that greatly affects the hydrogenation step during 2e⁻ ORR are investigated. Firstly, the $H_2O$/$D_2O$ kinetic isotope effect (KIE) experiment was conducted to explore the role of proton supply in $O_2$ hydrogenation to $H_2O_2$[51].

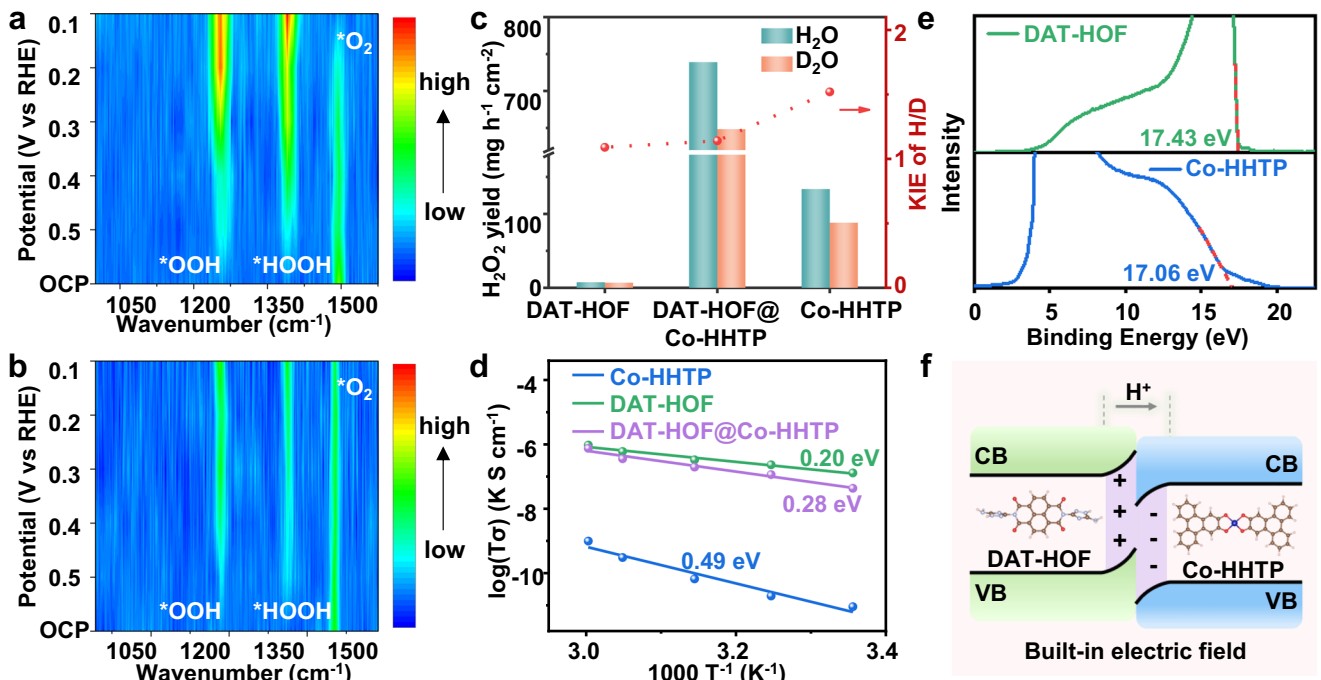

**Fig. 4 | Mechanistic studies.** Contour maps derived from the the in situ ATR-IR spectra recorded on **a** DAT-HOF@Co-HHTP and **b** Co-HHTP at different applied potentials. The color bars represent intensity with arb. units. **c** Arrhenius plots at different temperatures for DAT-HOF, Co-HHTP and DAT-HOF@Co-HHTP. **d** Kinetic isotope effect (KIE) of $H_2O/D_2O$ for ORR on DAT-HOF, Co-HHTP and DAT-HOF@Co-HHTP. **e** UPS spectra of DAT-HOF and Co-HHTP. **f** Built-in electric field in DAT-HOF@Co-HHTP. Source data for Fig. 4 are provided as a Source Data file.

Generally, the KIE value (the ratio of $H_2O_2$ yield in $H_2O$ and $D_2O$) closer to 1 indicates that the proton donation is not the rate-determining step (RDS). As shown in Fig. 4c, the KIE value of Co-HHTP is measured to be 1.52, indicating the inferior proton supply capability. After integrating DAF-HOF (KIE value of 1.09), the KIE value of DAT-HOF@Co-HHTP decreases to 1.14, showing enhanced proton supply for $H_2O_2$ production.

Subsequently, the proton conductivity (σ) of the samples was determined by alternating current impedance test under 95% relative humidity from 273 to 333 K[52]. As displayed in Supplementary Fig. 16, the σ value of DAT-HOF@Co-HHTP increases with temperature and reaches $6.64 \times 10^{-4}$ S cm$^{-1}$ at 333 K. The value is slightly lower than that of pristine DAT-HOF ($7.35 \times 10^{-4}$ S cm$^{-1}$), which can be attributed to the integration of Co-HHTP with a relatively lower proton conductivity (σ of $3.73 \times 10^{-6}$ S cm$^{-1}$). Based on the corresponding Arrhenius plots, the activation energies ($E_a$) in proton conduction are calculated[53]. According to literature, the proton transportation on solid materials follows two mechanisms: (1) the Grotthuss mechanism involves the proton diffusion via a H-bond network with $E_a$ between 0.1-0.4 eV[54]; (2) the vehicle mechanism describes the migration of protons coupled with movable carriers (e.g., $H_2O$) with $E_a > 0.4$ eV[55]. As shown in Fig. 4d, the $E_a$ value of Co-HHTP is calculated to be 0.49 eV, corresponding to a vehicular mechanism. In contrast, the proton transfer on DAT-HOF@Co-HHTP proceeds via the Grotthuss mechanism with a lower $E_a$ of 0.28 eV, which is mainly contributed by the core composition (DAT-HOF) with a H-bond network ($E_a$ of 0.2 eV)[56].

To elucidate the proton transfer at the DAT-HOF/Co-HHTP heterointerface, ultraviolet photoelectron spectroscopy (UPS) analysis was performed for determining the work function (Φ) of DAT-HOF and Co-HHTP using a monochromatic He light source (21.22 eV). The secondary electron cutoff energies ($E_{cutoff}$) of DAT-HOF and Co-HHTP are 17.43 and 17.06 eV (Fig. 4e), respectively. Based on the equation of $Φ = 21.22 - E_{cutoff}$, the Φ values of DAT-HOF and Co-HHTP were determined to be 3.79 and 4.16 eV (vs vacuum),

respectively[57]. According to the formula of $E_f = E_v - Φ$ ($E_v$ is the vacuum level, assumed as 0 eV), the Fermi levels ($E_f$) of DAT-HOF and Co-HHTP are calculated as −3.79 and −4.16 eV (vs vacuum). When DAT-HOF and Co-HHTP are in contact at the heterostructure, the difference in $E_f$ induces the spontaneous migration of free electrons from DAT-HOF to Co-HHTP until the $E_f$ equilibrium is established[58]. A built-in electric field directed from DAT-HOF to Co-HHTP is thus formed (Fig. 4f), offering driving force for promoting the proton transportation from DAT-HOF to Co-HHTP. Based on the results of in-situ ATR-IR, KIE, proton conductivity and UPS measurements, it is demonstrated that DAT-HOF with protonated structure serves as an efficient proton source, while the abundant hydrogen-bond networks provide highways for proton transfer with low energy barrier. Further contributed by the built-in electric field, DAT-HOF@Co-HHTP exhibits sufficient proton supply capability for $O_2$ hydrogenation toward $H_2O_2$ production.

Furthermore, density functional theory (DFT) simulation was conducted to gain deeper insights into the 2e$^-$ ORR process over DAT-HOF@Co-HHTP. The XPS observations suggest the interaction between DAT-HOF and Co-HHTP through cobalt-imine nitrogen coordination. Considering that there are two types of imine nitrogen in DAT-HOF including $N_1$ and $N_2$ as indicated in Supplementary Fig. 17a, the formation energies of two possible heterostructure models are firstly calculated (Supplementary Fig. 17b, c). The results (Supplementary Fig. 17 d) show that the formation energy of the heterostructure via Co-$N_1$ interaction (−0.96 eV) is more negative than the Co-$N_2$ counterpart (−0.79 eV). The DAT-HOF@Co-HHTP model with interfacial Co-$N_1$ bond is thus used for the subsequent calculations (Supplementary Data 1). The differential charge distribution diagram of the optimized heterostructure shows the electron accumulation around Co atom in Co-HHTP while electron depletion from N atom in DAF-HOF via the interfacial Co-N bridge (Fig. 5a), indicating the electron transfer from DAT-HOF to Co-HHTP, consistent with the XPS and UPS results.

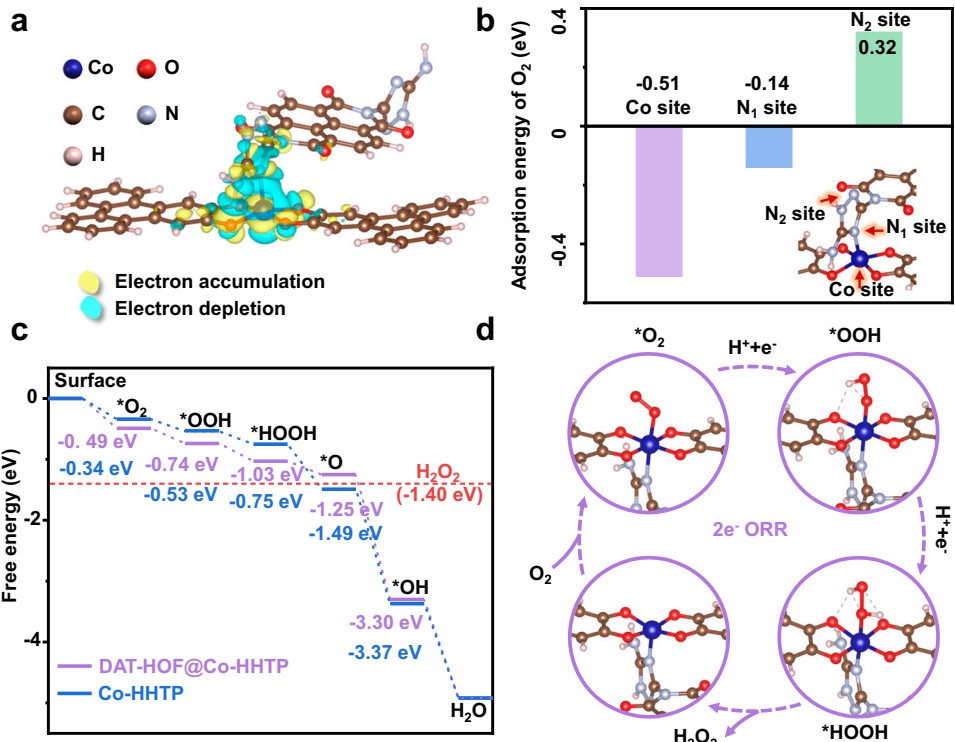

**Fig. 5 | DFT calculations. a** Calculated charge density difference of DAT-HOF@Co-HHTP heterostructure. **b** Calculated $O_2$ adsorption energies at the Co, $N_1$, and $N_2$ site of DAT-HOF@Co-HHTP, respectively. **c** Free energy diagrams for ORR over Co-HHTP and DAT-HOF@Co-HHTP heterostructure. **d** Schematic diagram of the ORR pathways over DAT-HOF@Co-HHTP heterostructure. Source data for Fig. 5 are provided as a Source Data file.

To identify the 2e⁻ ORR active site in DAT-HOF@Co-HHTP, the $O_2$ adsorption energies of different sites were calculated including $Co^{2+}$, $N_1$ and $N_2$ (Fig. 5b). The free energy change (ΔG) of $O_2$ adsorption on $Co^{2+}$ was −0.51 eV, lower than that on $N_1$ (−0.14 eV) and $N_2$ (0.32 eV), suggesting that the $Co^{2+}$ site is the active center for 2e⁻ ORR. Figure 5c, d depict the free-energy diagrams of 2e⁻ ORR and adsorption configurations of each step. The ΔG values of *$O_2$ for DAT-HOF@Co-HHTP and Co-HHTP are −0.49 and −0.34 eV, respectively, suggesting the enhanced $O_2$ adsorption and activation ability of DAT-HOF@Co-HHTP, in accordance with the observations of in-situ ATR-IR. For the second step of *$O_2$ to OOH*, a lower energy barrier is found for DAT-HOF@Co-HHTP (−0.25 eV) than Co-HHTP (−0.19 eV, Supplementary Data 2), indicating that the $O_2$ protonation over DAT-HOF@Co-HHTP is thermodynamically more favorable (Fig. 5c and Supplementary Fig. 18). For the second hydrogenation step from *OOH to *HOOH, the energy cost for DAT-HOF@Co-HHTP (0.22 eV) is also lower than that of Co-HHTP (0.29 eV), further underscoring the crucial role of DAT-HOF in facilitating proton supply. Moreover, the ΔG of *O formation via competitive 4e⁻ ORR pathway was calculated to be −0.51 eV for DAT-HOF@Co-HHTP, higher than that of $H_2O_2$ (−0.66 eV), indicating that its high 2e⁻ ORR selectivity has a thermodynamical origin. In contrast, the lower ΔG of $O_{ad}$ than $H_2O_2$ results in the low 2e⁻ ORR selectivity for Co-HHTP, consistent with the electrochemical results.

Collectively, the construction of DAT-HOF@Co-HHTP heterostructured electrocatalyst significantly improves the 2e⁻ ORR performance for $H_2O_2$ production. Within the heterostructure, DAT-HOF with a protonated structure and abundant H-bond networks serves as an efficient proton pump while Co-HHTP improves electron conductivity. Upon the contact of two components, the difference in Fermi levels drives the spontaneous electron transfer from DAT-HOF to Co-HHTP through the interfacial Co−N bonds. The produced electron-sufficient Co active sites with optimized binding strength of oxygen intermediates play a crucial role in enhancing the 2e⁻ ORR

activity and selectivity. Simultaneously, the charge redistribution results in the formation of an interfacial built-in electric field pointing from DAT-HOF to Co-HHTP, inducing the directional migration of protons from DAT-HOF to Co sites in Co-HHTP for $O_2$ hydrogenation to $H_2O_2$. By integrating the finely tuned active sites, high electron conductivity and sufficient proton supply capability, DAT-HOF@Co-HHTP is capable of driving ampere-level 2e⁻ ORR process with high performance.

## Discussion

In this work, a DAT-HOF@Co-HHTP heterostructured electrocatalyst has been synthesized for $H_2O_2$ electrosynthesis with a high $H_2O_2$ production rate of 738.9 mg h⁻¹ cm⁻² and a $H_2O_2$ FE of 97.1 ± 0.4% at a current density of 1200 mA cm⁻². The combination of Co-HHTP with DAT-HOF endows the heterostructure with high conductivity, sufficient proton supply and tailored active sites, synergistically promoting the 2e⁻ ORR to $H_2O_2$ production. Despite of the innovative HOF@c-MOF heterostructure design, our work has demonstrated the importance of sufficient proton supply towards ampere-level electrocatalytic production of $H_2O_2$ via the 2e⁻ ORR route. To realize practical industrial $H_2O_2$ electrosynthesis, further work should be conducted such as scalable synthesis of the developed electrocatalysts, developing optimized reactor systems, establishing effective product extraction strategies, and conducting a comprehensive techno-economic analysis to assess scalability and cost-competitiveness.

## Methods
### Reagents and materials
N, N-dimethylformamide (DMF, AR, 99%, Greagent), 1H-1,2,4-triazole-3,5-diamine (DAT, >98.0%, Adamas-beta), naphthalene-1,4,5,8-tetracarboxylic dianhydride (NTD, >97.0%, Adamas-beta), N,N-dimethylacetamide (DMA, 99%, Greagent), cobalt(II) acetate tetrahydrate (Co(OAc)₂·4H₂O, 99%, Adamas-beta), 2,3,6,7,10,11-hexahydroxytriphenylene (HHTP, 99%,

Adamas-beta), 1-propanol ($C_3H_8O$, ≥95.0%, Greagent) and ethanol ($C_2H_5OH$, 99%, Adamas-beta) were used as received. Millipore water was used in all experiments.

## Synthesis of DAT-HOF

DAT-HOF was synthesized according to the reported method[59]. Typically, 99.1 mg of DAT was dissolved into 10 mL of DMA by stirring in an ice bath under $N_2$ atmosphere for 30 min. Afterwards, 134 mg of NTD and 60 mL of DMA were added into the above mixture with stirring for another 30 min. The suspension was then transferred into a 100 mL Teflon-line autoclave and heated at 170 °C for 6 h. Finally, the resultant DAT-HOF was collected by centrifugation, washed by DMA and ethanol for three times and dried at 60 °C for further use.

## Synthesis of DAT-HOF@Co-HHTP

To synthesize DAT-HOF@Co-HHTP, 8 mg of DAT-HOF and 7 mg of HHTP was ultrasonically dispersed into a mixture solution containing 1 mL of DMA, 4 mL of 1-propanol and 4 mL of $H_2O$. Then, another mixture solution prepared by dissolving 6 mg of $Co(OAc)_2 \cdot 4H_2O$ in 4 mL of $H_2O$ and 4 mL of 1-propanol was poured into the above suspension. After heating at 55 °C for 1 h, the DAT-HOF@Co-HHTP powder was collected by centrifugation, washed with $H_2O$ and ethanol for three times, and dried at 60 °C.

## Synthesis of Co-HHTP

Co-HHTP was prepared by the similar process of DAT-HOF@Co-HHTP without the addition of DAT-HOF.

## Material characterization

Transmission electron microscopy (TEM) images were acquired by Hitachi HT7700 at 120 KV. Chemical composition analyses were carried out using JEM-2100F (JEOL, Japan) operating at 200 kV equipped with an X-ray energy dispersive spectrometer (EDS, X-Max 80 T, Oxford, UK). Field-emission scanning electron microscope (SEM) images were collected by scanning electron microscope (HITACHI-S4800). X-ray diffraction (XRD) patterns were recorded by a Bruker D8 Advanced X-Ray Diffractometer with Cu Kα radiation (λ = 0.154 nm). Fourier transform infrared (FTIR) spectra were collected on a Nicolet Fourier spectrophotometer using KBr pellets. X-ray photoelectron spectroscopy (XPS) studies were carried out on an AXIS Supra+ using an Al Kα radiation and C 1 s (284.8 eV) as a reference to correct the binding energy. The concentrations of metal ions were determined by Agilent 730 inductively coupled plasma-optical emission spectrometry (ICP-OES). UV-vis spectra were obtained by using a UV-vis spectrophotometer (Perkin Elmer Lambda 750). Ultraviolet photoelectron spectroscopy (UPS) studies were conducted by using an ESCALAB 250 XI analyzer with He (21.22 eV) as monochromatic light source. The X-ray absorption fine structure (XAFS) measurements were performed using the Rapid XAFS 1 M (Anhui Absorption Spectroscopy Analysis Instrument Co., Ltd.). Data analysis was performed with the Athena and Artemis programs of the Demeter data analysis packages that utilizes the FEFF6 program to fit the EXAFS data[60–62].

## Electrochemical measurement

**The ORR activities evaluated in rotating ring-disk electrode (RRDE) system.** Rotating ring-disk electrode (RRDE) tests were performed using a standard three-electrode system on a CHI-760C electrochemical workstation (CH Instruments Inc.) in $O_2$-saturated 0.1 M $K_2SO_4$ solution (200 mL, pH=7 ± 0.2) at ambient temperature (25 ± 1 °C). The electrolyte was freshly prepared for each measurement. Platinum wire (CH, CHI115), Ag/AgCl (KCl, 3.5 M, CHI111) and catalyst-modified glassy carbon were applied as the counter, reference and working electrodes, respectively. The Ag/AgCl reference electrode is calibrated using Potassium Ferrocyanide/Ferricyanide ($K_4[Fe(CN)_6]$/ $K_3[Fe(CN)_6]$) couple. The catalyst inks were prepared by dispersing

10 mg of sample into 1 mL of isopropanol containing 40 µL of Nafion solution to form a homogeneous suspension. The obtained ink was then dipped onto the polished glassy carbon disk (0.2475 cm$^{-2}$) under an infrared lamp. The loading amount of catalyst is determined to be 0.2 mg cm$^{-2}$. The RRDE was rotated at 1600 rpm throughout the whole tests. Liner sweep voltammetry (LSV) curves were recorded at a scan rate of 5 mV s$^{-1}$. A constant voltage of 1.2 V vs. RHE was applied to the ring electrode. The potential reaching the current density of 0.1 mA cm$^{-2}$ in polarization curves on disk electrode was defined as the onset potential. The selectivity of $H_2O_2$ was calculated using the following equation: Selectivity (%) = 200 × $(I_r/N)/(I_d+I_r/N)$. The electron transfer number (n) is calculated by the following equation:

$$n = 4 \times \frac{I_d}{I_d + I_d/N} \tag{1}$$

where $I_r$ is the ring current, $I_d$ is the disk current and N is the current collection efficiency of the Pt ring electrode (N = 0.256).

The Tafel slope (b) was obtained by fitting the linear part of the Tafel plots according to the Tafel equation

$$\eta = a + b\log(j) \tag{2}$$

to evaluate the kinetic performance of as-prepared catalysts for ORR. The electrochemical active surface area (ECSA) was evaluated based on the double-layer capacitances ($C_{dl}$) of the catalysts on RDE by cyclic voltammograms (CV) curves at different scanning rates of 10–100 mV s$^{-1}$ in the non-Faradaic voltage region. A straight line can be obtained by plotting the current density against the scan rate at a specific potential in the CV curves. The slope of the line is defined as electrochemical double-layer capacitance ($C_{dl}$). Furthermore, the ECSA can be calculated as:

$$ECSA = \frac{C_{dl}}{A \times Cs} \tag{3}$$

where A is the amount of the material coating on the surface of electrode (mg cm$^{-2}$), Cs is an empirical constant representing the capacitance per unit area (40 µF cm$^{-2}$). Electrochemical impedance spectroscopy (EIS) was measured in 0.1 M $K_2SO_4$ solution in the frequency range of 1000 kHz to 0.01 Hz with an amplitude of 10 mV. All the potentials were calibrated with a reversible hydrogen electrode (RHE)

$$E_{RHE} = E_{Ag/AgCl} + 0.0591 \times pH + 0.197 \tag{4}$$

All the electrochemical tests were performed without IR compensation.

## Electrochemical evaluation in the flow-type cell

Electrochemical tests were carried out on a CHI760E electrochemical workstation connected to a CHI680D high current amplifier at ambient temperature (25 ± 1 °C). A standard three-electrode three-phase flow cell system was assembled by employing gas diffusion electrode (GDE) as a working electrode, platinum electrode as a counter electrode, and Ag/AgCl (KCl, 3.5 M, CHI111) as a reference electrode. The catalyst inks were prepared by dispersing 10 mg of sample into 1 mL of isopropanol containing 40 µL of Nafion solution to form a homogeneous suspension. Next, the homogeneous catalyst ink was dripped on GDE (chuxi) with an overall area of 2 × 2 cm$^2$ (active area of 1 × 1 cm$^2$). The loading amount of catalyst is determined to be 0.2 mg cm$^{-2}$. The catholyte and anolyte were both 0.1 M $K_2SO_4$ aqueous solution (500 mL, pH=7 ± 0.2). The electrolyte was freshly prepared for each measurement. The electrolytic cell is separated by an anion exchange membrane

(FunasepFAA-3-PK-130, 130 μm, 1 × 1 cm²). The membrane is pretreated by immersion in a 0.5 M NaCl solution at 25 °C for 24 h. A peristaltic pump was used to circulate the electrolyte with a rotational speed of 30 rpm. The $O_2$ gas flow was maintained as 20 mL min⁻¹ during the whole measurement. The $H_2O_2$ production rate was determined by the iodometry method. Typically, 100 μL of reaction solution was collected from the electrochemical system and subsequently added to the mixture of potassium hydrogen phthalate ($C_8H_5KO_4$) and potassium iodide (KI) aqueous solution with reaction for 30 min. The $H_2O_2$ were allowed to react with I⁻ to generate $I_3^-$ ($H_2O_2 + 3I^- + 2H^+ \rightarrow I_3^- + H_2O$). The amount of $I_3^-$ was measured by a Synergy-H1 microplate reader at its characteristic absorbance peak of 350 nm for $H_2O_2$ quantification. All the electrochemical tests were performed without IR compensation.

## Computational details

The Density Functional Theory (DFT) calculations were conducted using the Vienna Ab-inito Simulation Package (VASP)[63,64]. The Perdew-Burke-Ernzerhof (PBE) with the generalized gradient approximation (GGA) method was employed to describe the exchange-correlation effects[65,66]. The core-valence interactions were calculated by the projected augmented wave (PAW) method[67]. An energy cutoff of 500 eV was set for plane wave expansions, and the 3 × 2 × 1 Monkhorst-Pack grid k-points were selected to sample the Brillouin zone integration. The vacuum space is adopted 15 Å above the surfaces to avoid periodic interactions. The structural optimization was completed for energy and force convergence set at 1.0 × 10⁻⁴ eV and 0.02 eV Å⁻¹, respectively. The models were constructed based on the standard crystalline structure of DAT-HOF and Co-HHTP.

The adsorption energy can be calculated according to the following formula:

$$Eads = E(A + B) - E(A) - E(B) \qquad (5)$$

where Eads represents the adsorption energy, E(A + B) is the calculated energy of adsorption configuration, E(A) and E(B) mean the calculated energy of substrate and adsorbent respectively.

The Gibbs free energy change (ΔG) of each step is calculated using the following formula:

$$\Delta G = \Delta E + \Delta ZPE - T\Delta S \qquad (6)$$

where ΔE is the electronic energy difference directly obtained from DFT calculations, ΔZPE is the zero point energy difference, T is the room temperature (298.15 K) and ΔS is the entropy change. ZPE could be obtained after frequency calculation by ref. 68:

$$ZPE = \frac{1}{2}\sum h\nu i \qquad (7)$$

The TS values of adsorbed species are calculated according to the vibrational frequencies[69]:

$$TS = k_B T \left[ \sum_k \ln\left(\frac{1}{1 - e^{-h\nu/k_B T}}\right) + \sum_k \frac{h\nu}{k_B T}\frac{1}{(e^{h\nu/k_B T} - 1)} + 1 \right] \qquad (8)$$

## Data availability

The raw data generated in this study are provided in the Supplementary Information. All data are available from the corresponding author upon request. Source data are provided with this paper Source data are provided with this paper.

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

## Acknowledgements

The authors acknowledge support from the National Natural Science Foundation of China (Grant No. 22572056, 22075085, and 22475072), the Natural Science Foundation of Shanghai (25ZR1402122), the Fundamental Research Funds for the Central Universities, the Shanghai Frontiers Science Center of Molecule Intelligent Syntheses, East China Normal University (ECNU) Multifunctional Platform for Innovation (004) and the Supercomputer Center of East China Normal University (ECNU Multifunctional Platform for Innovation 001).

## Author contributions

Y.Z. (Yingying Zou) and C.L. conceived and supervised the project. Y.Z. (Yingying Zou) and Y.Z. (Yulin Zhang) performed the majority of material synthesis, characterization, and electrochemical measurements. C.Z., T.B., Y.X. and N.A. contributed to electrochemical measurements and scientific discussions. C.Z. carried out the DFT calculations. The manuscript was written by Y.Z., C.L. and C.Y. with support from all co-authors.

## Competing interests

The authors declare no competing interests.
