## [Transparent Peer Review file · Nature Communications]

Hydrogen-Bonded Organic Framework@Conductive Metal-Organic Framework Heterostructures for Ampere-Level Hydrogen Peroxide Production

Corresponding Author: Professor Chengzhong Yu

Version 0:

Reviewer comments:

Reviewer #1

(Remarks to the Author)

This study presents a promising approach for developing high-performance $2e^-$ ORR electrocatalysts by constructing HOF@cMOF heterostructures, achieving an impressive current density of 1.2 A/cm^2 (even up to 1.4 A/cm^2 in some cases). Obvious self-contradictory discussion happens to the main claims on the effect of proton and electron conduction in this work. To fully support their claims, the authors should address the following issues carefully. Otherwise, it will not be considered for possible publication Nature communications.

1. The mechanism proposed in Scheme 1, which attributes the enhanced ORR activity primarily to bulk proton and electron conduction, appears inconsistent with the experimental evidence. The observed activity trend (DAT-HOF@Co-HHTP > Co-HHTP > DAT-HOF) reveals several critical contradictions: (1) XAS and XPS data demonstrate electron enrichment at Co sites in DAT-HOF@Co-HHTP compared to Co-HHTP, suggesting charge transfer-induced electronic modulation plays a more significant role than bulk conduction; (2) The higher ECSA-normalized activity of DAT-HOF@Co-HHTP indicates superior intrinsic activity of the catalytic sites; (3) Despite exhibiting lower bulk conductivity than Co-HHTP, DAT-HOF@Co-HHTP shows significantly enhanced activity, implying electron conduction is not the limiting factor; (4) While DAT-HOF@Co-HHTP displays lower proton conductivity than pure DAT-HOF, its superior performance suggests bulk proton transport is not the dominant factor - though localized proton donation at active sites may contribute. Although proton conduction might become relevant when comparing systems with similar electronic states and intrinsic activities, the current data cannot conclusively support proton/electron conduction as the primary drivers of enhanced performance.

These inconsistencies indicate that Scheme 1 oversimplifies the actual mechanism. The authors should respect the experimental result and reconsider their interpretation to better account for the dominant role of electronic structure modulation and interfacial effects in enhancing the intrinsic activity of Co sites, rather than focusing solely on bulk conduction properties. While proton and electron conduction may indeed influence ORR activity, the presented results do not provide sufficient evidence to establish these factors as the key determinants of the observed performance enhancement. Based on this, the logic of the introduction part should be revised accordingly.

2. The reported current density is remarkable, but key experimental information is missing: (i) Ink composition: Does it include Nafion (proton conductor) and carbon black (electron conductor)? (ii) Electrode preparation: How was the gas diffusion layer (GDL) fabricated? (iii) Electrolyte conditions: The electrolyte composition and pH should be specified. These details are essential for reproducibility and should be clearly stated in either the main text or Supporting Information.

3. In the flow cell measurement part (concerning the high current density), some important data is missing. Linear sweep voltammetry (LSV) data in the flow cell system should be provided to substantiate the high current density claims. Chronoamperometry (I-t) or potential-time (V-t) curves are needed to demonstrate long-term stability.

4. Could the authors clarify the crystal structure of DAT-HOF? Is the single-crystal structure available for analysis? If not, has the structure been simulated or refined from the PXRD pattern using methods like Pawley refinement or Rietveld analysis?

5. I think the main XRD peak for Co-HHTP (Figure 1d) is missing, please give the pattern start from 4θ (2 theta).

6. To confirm the successful synthesis of DAT-HOF and Co-HHTP (rather than mere molecular packing), physical sorption experiments (e.g., N_2/Ar adsorption) should be provided to verify their porosity.

7. The overlay of Co (MOF) and N (HOF) would better demonstrate the heterostructure formation, rather than C/O/Co/N, C and O are common to both components.

8. Formation and data Presentation Issues. (i) Figure labeling: Samples in Figure 2b and Figure S8 should be clearly labeled. (ii) Missing captions: Figure 3e and 3f lack proper explanations. (iii) Typographical errors: E.g., Line 166: "shift of

@@ A compared to Co-HHTP" should be corrected. A thorough proofreading is recommend

Reviewer #2

(Remarks to the Author)

In this work, a hydrogen-bonded organic framework@conductive metal-organic framework (HOF@cMOF) heterostructure is designed for industrial-level H₂O₂ electrosynthesis. The heterostructure is constructed by integrating DAT-HOF with high proton donation capability and Co-HHTP with high conductivity. The Co-N bonds at the heterointerface modulates the electronic structure of Co active sites and the formation of built-in electric field drives the directional proton migration from DAT-HOF to Co-HHTP, facilitating the O₂ protonation to H₂O₂ at Co sites. Efficient H₂O₂ production under ampere-level current densities is achieved with a H₂O₂ Faradic efficiency of 97.1%, a H₂O₂ yield of 738.9 mg h⁻¹ cm⁻² and a long-term durability for over 100 h at 1200 mA cm⁻². This work is well organized and can be considered as a publication after addressing the following issues.

1. There are some grammar issues in this work, such as on the page 8, line 166 ("For DAT-HOF@Co-HHTP, the peak exhibits a slight negative shift of @@ Å compared to Co-HHTP, possibly resulted by the formation of Co-N bond at DAT-HOF/Co-HHTP interface").
2. In Figure 2g, the XRD pattern of extracted solid H₂O₂ is not completely consistent with the PDF#11-0656.
3. How to demonstrate the intermolecular hydrogen bonding between DAT-HOF@Co-HHTP.
4. On page 12, line 262-266, "As displayed in Fig.S10, the σ value of DAT-HOF@Co-HHTP increases with temperature and reaches 6.64×10^{-4} S cm⁻¹ at 333 K, slightly lower than DAT-HOF (7.35×10^{-4} S cm⁻¹) but significantly higher than Co-HHTP (3.73×10^{-6} S cm⁻¹). Why the σ of DAT-HOF@Co-HHTP is lower.

Reviewer #3

(Remarks to the Author)

This manuscript described the synthesis of a heterostructured electrocatalyst by integrating DAT-HOF with high proton donation capability and Co-HHTP with high conductivity for ampere-level hydrogen peroxide production. The obtained DAF-HOF/Co-HHTP electrocatalyst delivered a superior 2e⁻ ORR performance with a H₂O₂ Faradic efficiency of 97.1%, a H₂O₂ yield of 738.9 mg h⁻¹ cm⁻² and a long-term durability for over 100 h at 1200 mA cm⁻². Systematical investigations revealed that the adsorption strength of key oxygen intermediates on the Co active sites was optimized by forming Co-N chemical bonds at the heterointerface. In addition, a built-in electric field was formed for accelerating proton migration from DAT-HOF to Co-HHTP, accelerating the protonation kinetics of 2e⁻ ORR. The material design of HOF@conductive MOFs heterostructure is innovative and the performance is impressive. Overall, the manuscript is recommended for publication pending minor revisions as detailed below.

- (1) The experimental conditions of flow cell test are not specified.
- (2) The figure captions for Figure. 3e-f are missing.
- (3) The electron transfer numbers of the three catalysts should be provided.
- (4) In Figure 2e, the H₂O₂ yield and FE of DAT-HOF@Co-HHTP heterostructure were compared with those of reported electrocatalysts. The detailed reaction conditions of both literature and this work should be listed in a table in the supporting information.
- (5) The DFT calculations should also involve the *OOH → *HOOH hydrogenation step to further support the facilitated hydrogenation by integrating HOF.

Version 1:

Reviewer comments:

Reviewer #1

(Remarks to the Author)

Manuscript Number: NCOMMS-25-53783

I am satisfied with the revised manuscript and recommend its acceptance for publication as it is.

Reviewer #2

(Remarks to the Author)

The authors have revised the manuscript accordingly. It can be considered as a publication in the present form.

Reviewer #3

(Remarks to the Author)

My comments in the last round of review have been well addressed. I believe it can now be published as it is.

Point-to-point response and revisions

(Original Manuscript Number: NCOMMS-25-53783)

Reviewer 1#

This study presents a promising approach for developing high-performance $2e^-$ ORR electrocatalysts by constructing HOF@cMOF heterostructures, achieving an impressive current density of 1.2 A/cm^2 (even up to 1.4 A/cm^2 in some cases). Obvious self-contradictory discussion happens to the main claims on the effect of proton and electron conduction in this work. To fully support their claims, the authors should address the following issues carefully. Otherwise, it will not be considered for possible publication Nature communications.

Response: We thank Reviewer 1 for the positive comment stating that “This study presents a promising approach for developing high-performance $2e^-$ ORR electrocatalysts by constructing HOF@cMOF heterostructures, achieving an impressive current density of 1.2 A/cm^2 (even up to 1.4 A/cm^2 in some cases)”. Your insightful queries have encouraged us to rereview the key determinants of the performance enhancement. We have reorganized the Scheme 1 and discussions in related sections (e.g., abstract, introduction, results and discussion) to highlight the predominated role of electronic structure modulation and interfacial effects in enhancing the intrinsic activity of Co sites (see details in our response of your comment 2). In addition, your other comments are also highly valuable and important for us to improve the quality of our work. In the revised manuscript, we have conducted additional experiments with further revisions to address your queries (see details below).

Comment 1. The mechanism proposed in Scheme 1, which attributes the enhanced ORR activity primarily to bulk proton and electron conduction, appears inconsistent with the experimental evidence. The observed activity trend (DAT-HOF@Co-HHTP >

Co-HHTTP > DAT-HOF) reveals several critical contradictions: (1) XAS and XPS data demonstrate electron enrichment at Co sites in DAT-HOF@Co-HHTTP compared to Co-HHTTP, suggesting charge transfer-induced electronic modulation plays a more significant role than bulk conduction; (2) The higher ECSA-normalized activity of DAT-HOF@Co-HHTTP indicates superior intrinsic activity of the catalytic sites; (3) Despite exhibiting lower bulk conductivity than Co-HHTTP, DAT-HOF@Co-HHTTP shows significantly enhanced activity, implying electron conduction is not the limiting factor; (4) While DAT-HOF@Co-HHTTP displays lower proton conductivity than pure DAT-HOF, its superior performance suggests bulk proton transport is not the dominant factor - though localized proton donation at active sites may contribute. Although proton conduction might become relevant when comparing systems with similar electronic states and intrinsic activities, the current data cannot conclusively support proton/electron conduction as the primary drivers of enhanced performance.

These inconsistencies indicate that Scheme 1 oversimplifies the actual mechanism. The authors should respect the experimental result and reconsider their interpretation to better account for the dominant role of electronic structure modulation and interfacial effects in enhancing the intrinsic activity of Co sites, rather than focusing solely on bulk conduction properties. While proton and electron conduction may indeed influence ORR activity, the presented results do not provide sufficient evidence to establish these factors as the key determinants of the observed performance enhancement. Based on this, the logic of the introduction part should be revised accordingly.

Response: We sincerely thank Reviewer 1 for the constructive and insightful comments. In the revised manuscript, the following revisions have been made to highlight the predominate role of electronic structure modulation and interfacial effects in enhancing the intrinsic activity of Co sites.

(1) The abstract has been modified as follows:

"Electrochemical two-electron oxygen reduction reaction ($2e^-$ ORR) in neutral environments holds remarkable promise for sustainable hydrogen peroxide (H_2O_2)

production. However, its practical application is largely hindered due to the scarcity of electrocatalysts with high selectivity and durability under ampere-level current densities. Herein, a hydrogen-bonded organic framework@conductive metal-organic framework (HOF@cMOF) heterostructure is designed for industrial-level H₂O₂ electrosynthesis. Through the integration of DAT-HOF (DAT=diaminotriazole) and Co-HHTP (HHTP=2,3,6,7,10,11-hexahydroxytriphenylene), the formation of Co-N bonds at the heterointerface modulates the electronic structure of Co active sites, optimizing the adsorption strength of oxygen intermediates toward improved 2e⁻ ORR activity and selectivity. Besides, the formation of built-in electric field drives the directional proton migration from DAT-HOF to Co-HHTP, facilitating the O₂ protonation to H₂O₂ at Co sites. In further combination with the high proton donation capability of DAT-HOF and high conductivity of Co-HHTP, efficient H₂O₂ production under ampere-level current densities is achieved with a H₂O₂ Faradic efficiency of 97.1%, a H₂O₂ yield of 738.9 mg h⁻¹ cm⁻² and a long-term durability for over 100 h at 1200 mA cm⁻². This work offers a high-performance electrocatalyst for promoting the industrial implementation of H₂O₂ electrosynthesis."

(2) The Scheme 1 has been revised as below.

Scheme 1. Schematic illustration of the (a) synthesis process, (b) interfacial active site and (c) built-in electric field of DAT-HOF@Co-HHTP heterostructure.

(3) The following revisions have been made in the introduction section:

"The precise modulation of catalytic active sites to suppress the $4e^-$ ORR pathway is essential for the selective production of H_2O_2 ."

"Further, the versatile surface functional groups of HOFs enable effective hybridization with other materials, yielding heterostructures with synergistic properties.³⁹ More importantly, the chemical interaction at the heterointerface can induce charge redistribution and modulate the electronic structure of catalytic metal sites, thereby enhancing the intrinsic activity and selectivity.⁴⁰ In addition, the abundant hydrogen-bond networks in HOFs can facilitate proton conduction, potentially enhancing local proton donation during electrocatalysis.⁴¹"

"At the HOF-*c*MOF heterointerface, the formation of Co-N bonds results in the electron-enriched Co active sites with optimized adsorption strength of oxygen intermediates, making pivotal contributions to the enhanced $2e^-$ ORR activity and selectivity. In addition, the charge redistribution between two components with different Fermi levels induces the formation of a built-in electric field, driving the directional migration of protons from DAT-HOF to Co-HHTP for facilitating the protonation of O_2 to H_2O_2 at Co sites. By integrating modulated active sites, high electron conductivity and enhanced hydrogenation capability, the rationally designed heterostructure delivers the remarkable $2e^-$ ORR performance at ampere-level current densities, superior than most reported electrocatalysts."

(4) The descriptions of the catalytic mechanism in the results and discussion part have also been revised as follows:

"Upon the contact of two components, the difference in Fermi levels drives the spontaneous electron transfer from DAT-HOF to Co-HHTP through the interfacial Co-N bonds. The produced electron-sufficient Co active sites with optimized binding strength of oxygen intermediates play a crucial role in enhancing the $2e^-$ ORR activity

and selectivity. Simultaneously, the charge redistribution results in the formation of an interfacial built-in electric field pointing from DAT-HOF to Co-HHTP, inducing the directional migration of protons from DAT-HOF to Co sites in Co-HHTP for O₂ hydrogenation to H₂O₂. By integrating the finely tuned active sites, high electron conductivity and sufficient proton supply capability, DAT-HOF@Co-HHTP is capable of driving ampere-level 2e⁻ ORR process with high performance."

Comment 2. The reported current density is remarkable, but key experimental information is missing: (i) Ink composition: Does it include Nafion (proton conductor) and carbon black (electron conductor)? (ii) Electrode preparation: How was the gas diffusion layer (GDL) fabricated? (iii) Electrolyte conditions: The electrolyte composition and pH should be specified. These details are essential for reproducibility and should be clearly stated in either the main text or Supporting Information.

Response: We sincerely appreciate Reviewer 1's helpful comments. The experiment information has been provided in the revised manuscript as follows:

"4.2 Electrochemical evaluation in the flow-type cell

Electrochemical tests were carried out on a CHI760E electrochemical workstation connected to a CHI680D high current amplifier. A standard three-electrode three-phase flow cell system was assembled by employing gas diffusion electrode (GDE) as a working electrode, platinum electrode as a counter electrode, and Ag/AgCl (KCl, 3.5 M) as a reference electrode. The catalyst inks were prepared by dispersing 10 mg of sample into 1 mL of isopropanol containing 40 μL of Nafion solution to form a homogeneous suspension. Next, 50 μL of the homogeneous catalyst ink was dripped on GDE with an overall area of 2 × 2 cm² and active area of 1 × 1 cm²). The catholyte and anolyte were both 0.1 M K₂SO₄ aqueous solution (pH=7). A peristaltic pump was used to circulate the electrolyte with a rotational speed of 30 rpm. The O₂ gas flow was maintained as 20 mL min⁻¹ during the whole measurement. The H₂O₂ production rate was determined by the iodometry method. Typically, 100 μL of reaction solution was collected from the electrochemical system and subsequently added to the mixture of

potassium hydrogen phthalate (C₈H₅KO₄) and potassium iodide (KI) aqueous solution with reaction for 30 min. The H₂O₂ were allowed to react with I⁻ to generate I₃⁻ (H₂O₂ + 3I⁻ + 2H⁺ → I₃⁻ + H₂O). The amount of I₃⁻ was measured by a Synergy-H1 microplate reader at its characteristic absorbance peak of 350 nm for H₂O₂ quantification."

Comment 3. In the flow cell measurement part (concerning the high current density), some important data is missing. Linear sweep voltammetry (LSV) data in the flow cell system should be provided to substantiate the high current density claims. Chronoamperometry (I-t) or potential-time (V-t) curves are needed to demonstrate long-term stability.

Response: We thank Reviewer 1 for the professional comments. The linear sweep voltammetry (LSV) and chronoamperometry (I-t) curves collected in the flow cell have been added as Figure S8 and S9 in the revised manuscript as below.

Figure S8. LSV polarization curves of DAT-HOF@Co-HHTP, DAT-HOF and Co-HHTP in flow cell.

Figure S9. Durability test under -3.9 V vs RHE for DAT-HOF@Co-HHTP.

The following descriptions have been added in the revised manuscript.

"Moreover, the LSV polarization curves measured in the flow electrolytic cell system (Fig. S8) show that DAT-HOF@Co-HHTP exhibits increased ORR current density and activity than DAT-HOF and Co-HHTP, consistent with the results obtained from the RRDE measurements."

"Additionally, the chronoamperometry (I-t) curve shows a nearly constant current density during a long-period operation of 100 h (Fig. S9), further demonstrating the excellent electrochemical stability of DAT-HOF@Co-HHTP."

Comment 4. Could the authors clarify the crystal structure of DAT-HOF? Is the single-crystal structure available for analysis? If not, has the structure been simulated or refined from the PXRD pattern using methods like Pawley refinement or Rietveld analysis?

Response: We thank Reviewer 1 for this insightful question. In order to obtain large-size single crystals, we have extended the reaction time from 6 h to two weeks. However, the resultant sample still exhibits relatively small particle size (< 2 μm). Therefore, according to your suggestion, we have performed Pawley refinement of the PXRD pattern to further clarify the crystal structure of DAT-HOF. The results and corresponding descriptions have been added in the revised manuscript as below.

Figure R1. SEM image of DAT-HOF obtained after two-week reaction time.

Figure S1c. Experimental (red dot) and refined (black line) PXR D patterns of DAT-HOF.

The following descriptions have been added in Page 5 in the revised manuscript.

"The crystalline structure of DAT-HOF was studied by powder X-ray diffraction (PXR D). As shown in Fig. S1c, the experimental PXR D pattern matches well with the simulated profile via Pawley refinement ($R_p = 5.26\%$ and $R_{wp} = 6.76\%$), revealing a hexagonal crystal structure with cell parameters of $a = 14.25$, $b = 14.25$ Å, $c = 5.17$ Å."

Comment 5. I think the main XRD peak for Co-HHTP (Figure 1d) is missing, please give the pattern start from 40 (2 theta).

Response: We acknowledge Reviewer 1's valuable suggestion. The XRD patterns over an extended range ($2\theta = 3-80^\circ$) have been provided in the revised manuscript as below.

Figure 1. (f) XRD patterns of DAT-HOF@Co-HHTP, DAT-HOF and Co-HHTP.

Figure S2. (c) XRD pattern of Co-HHTP.

Comment 6. To confirm the successful synthesis of DAT-HOF and Co-HHTP (rather than mere molecular packing), physical sorption experiments (e.g., N_2/Ar adsorption) should be provided to verify their porosity.

Response: We thank Reviewer 1 for raising this important point. As suggested, we have performed N_2 sorption measurements at 77 K for DAT-HOF, Co-HHTP, and DAT-HOF@Co-HHTP. The results are provided as Figure S5 in the revised manuscript, see also below.

Fig. S5. Nitrogen adsorption-desorption isotherms of DAT-HOF, Co-HHTP and DAT-HOF@Co-HHTP.

The following descriptions have been added in the revised manuscript.

"Figure S5 shows the N₂ adsorption-desorption isotherms of the samples. The Brunauer-Emmett-Teller (BET) specific surface areas and pore volumes of DAT-HOF, Co-HHTP and DAT-HOF@Co-HHTP were determined to be 75.1 m² g⁻¹ and 0.29 cm³ g⁻¹ for DAT-HOF, 91.9 m² g⁻¹ and 0.46 cm³ g⁻¹ for DAT-HOF@Co-HHTP, and 118.3 m² g⁻¹ and 0.50 cm³ g⁻¹ for Co-HHTP, respectively."

7. The overlay of Co (MOF) and N (HOF) would better demonstrate the heterostructure formation, rather than C/O/Co/N, C and O are common to both components.

Response: We thank Reviewer 1 for the useful suggestion. The overlay of Co (MOF) and N (HOF) has been provided in the revised manuscript as below.

Comment 8. Formation and data Presentation Issues. (i) Figure labeling: Samples in Figure 2b and Figure S8 should be clearly labeled. (ii) Missing captions: Figure 3e and 3f lack proper explanations. (iii) Typographical errors: E.g., Line 166: “shift of @@ Å compared to Co-HHTP” should be corrected. A thorough proofreading is recommended.

Response: We thank Reviewer 1 for the useful suggestions. We have carefully revised the manuscript as follows.

(i) Samples in Figure 2b and Figure S8 (Figure S13 in revised manuscript) have been clearly labeled.

Figure 2b. H₂O₂ selectivity of DAT-HOF@Co-HHTP, DAT-HOF and Co-HHTP.

Figure S13. EIS spectra of DAT-HOF, Co-HHTP and DAT-HOF@Co-HHTP.

(ii) The captions of Figure 3e and 3f have been added.

Figure 3. (e) UPS spectra of DAT-HOF and Co-HHTP. (f) Built-in electric field in DAT-HOF@Co-HHTP.

(iii) Typographical errors have been revised.

The sentence “shift of @@ Å compared to Co-HHTP” has been replaced by “shift of 0.32 Å compared to Co-HHTP” in Fig. 3.

Reviewer #2

In this work, a hydrogen-bonded organic framework@conductive metal-organic framework (HOF@cMOF) heterostructure is designed for industrial-level H₂O₂ electrosynthesis. The heterostructure is constructed by integrating DAT-HOF with high proton donation capability and Co-HHTP with high conductivity. The Co-N bonds at the heterointerface modulates the electronic structure of Co active sites and the formation of built-in electric field drives the directional proton migration from DAT-HOF to Co-HHTP, facilitating the O₂ protonation to H₂O₂ at Co sites. Efficient H₂O₂ production under ampere-level current densities is achieved with a H₂O₂ Faradic efficiency of 97.1%, a H₂O₂ yield of 738.9 mg h⁻¹ cm⁻² and a long-term durability for over 100 h at 1200 mA cm⁻². This work is well organized and can be considered as a publication after addressing the following issues.

Response: We thank Reviewer 2 for the positive comments.

Comment 1. There are some grammar issues in this work, such as on the page 8, line 166 (“For DAT-HOF@Co-HHTP, the peak exhibits a slight negative shift of @@ Å compared to Co-HHTP, possibly resulted by the formation of Co-N bond at DAT-HOF/Co-HHTP interface”).

Response: We thank Reviewer 2 for pointing out our mistakes, which have been revised as follows:

"For DAT-HOF@Co-HHTP, the peak exhibits a slight negative shift of 0.32 Å compared to Co-HHTP, possibly resulted from the formation of Co-N bond at DAT-HOF/Co-HHTP interface"

Comment 2. In Figure 2g, the XRD pattern of extracted solid H₂O₂ is not completely consistent with the PDF#11-0656.

Response: We are grateful for Reviewer 2's valuable comment. We have carefully compared the XRD patterns of solid H₂O₂ and standard Na₂CO₃·1.5H₂O₂, and found that the peak positions of solid H₂O₂ well match those of the standard pattern (Figure 2, see below). The variation in relative peak intensity may be attributed to the difference in preferred orientation between solid and standard sample. To further verify the successful preparation of solid H₂O₂, the Raman spectra of solid H₂O₂ and standard Na₂CO₃·1.5H₂O₂ have been added as Figure S11 in the revised manuscript, showing almost identical Raman peaks.

Figure 2. (g) XRD pattern of extracted solid H₂O₂ (insets are photographs of solid H₂O₂ and degradation of different dyes by solid H₂O₂).

Figure S11. Raman spectra of extracted solid H₂O₂ and standard Na₂CO₃·1.5H₂O₂.

The following descriptions have been added in the revised manuscript (page 10)
"By further adding Na₂CO₃ into the electrolyte after a successive electrolysis of 100 h, the produced H₂O₂ can be extracted in the form of sodium carbonate perhydrate (Na₂CO₃·1.5H₂O₂) as a solid H₂O₂ product. The XRD pattern of collected powder shows well-matched diffraction peaks with those of standard pattern, verifying the formation of solid H₂O₂ (Fig. 2g).⁴⁸ The variation in relative peak intensity may be attributed to the difference in preferred orientation between solid and standard sample. Additionally, the Raman spectra of solid H₂O₂ and standard Na₂CO₃·1.5H₂O₂ (Fig. S11) show almost identical patterns, further verifying the successful preparation of solid H₂O₂."

Comment 3. How to demonstrate the intermolecular hydrogen bonding between DAT-HOF@Co-HHTP.

Response: We thank Reviewer 2 for this thoughtful comment. Within DAT-HOF@Co-HHTP heterostructure, the two components are coordinatively interacted via Co–N bond between Co in Co-HHTP and N in DAT-HOF, as evidenced by the XAS measurements and DFT calculations. Even so, your valuable suggestion promotes us to further probe the presence of H-bond network via FTIR analysis. As shown in the FTIR spectra of DAT, NTD, DAT-HOF and DAT-HOF@Co-HHTP (Fig. S4, see below), the peaks at 3251 and 3070 cm⁻¹ are associated with the stretching vibrations of H-bond N–H, indicating the presence of H-bond network (Ref. 10.1002/anie.202506892, 10.1002/smll.202309353).

Figure S4. FTIR spectra of DAT-HOF@Co-HHTP, DAT-HOF and Co-HHTP.

The relevant descriptions have been incorporated into the revised manuscript as follows: "Compared to DAT and NTD ligands, the additional peaks at 3247 cm^{-1} and 3066 cm^{-1} in the spectrum of DAT-HOF are associated with the stretching vibration of H-bond N-H (Fig. S4), indicating the formation of hydrogen bonding in DAT-HOF.⁴⁵⁻⁴⁶ For Co-HHTP, the peaks at 1454 , 1298 and 1216 cm^{-1} in the spectrum are assigned to the benzene skeleton vibrations of HHTP.⁴⁷ By integrating DAT-HOF and Co-HHTP, the typical peaks of two components are simultaneously detected in DAT-HOF@Co-HHTP."

Comment 4. On page 12, line 262-266, "As displayed in Fig.S10, the σ value of DAT-HOF@Co-HHTP increases with temperature and reaches $6.64 \times 10^{-4}\text{ S cm}^{-1}$ at 333 K, slightly lower than DAT-HOF ($7.35 \times 10^{-4}\text{ S cm}^{-1}$) but significantly higher than Co-HHTP ($3.73 \times 10^{-6}\text{ S cm}^{-1}$). Why the σ of DAT-HOF@Co-HHTP is lower.

Response: We sincerely thank the reviewer for this insightful comment. The lower σ of DAT-HOF@Co-HHTP than DAT-HOF is resulted from the integration of Co-HHTP with a relatively lower proton conductivity (σ of $3.73 \times 10^{-6}\text{ S cm}^{-1}$). Further

clarifications have been added in Page 12 in the revised manuscript as follows:

"As displayed in Fig. S15, the σ value of DAT-HOF@Co-HHTP increases with temperature and reaches $6.64 \times 10^{-4} \text{ S cm}^{-1}$ at 333 K. The value is slightly lower than that of pristine DAT-HOF ($7.35 \times 10^{-4} \text{ S cm}^{-1}$), which can be attributed to the integration of Co-HHTP with a relatively lower proton conductivity (σ of $3.73 \times 10^{-6} \text{ S cm}^{-1}$)."

Reviewer #3

This manuscript described the synthesis of a heterostructured electrocatalyst by integrating DAT-HOF with high proton donation capability and Co-HHTP with high conductivity for ampere-level hydrogen peroxide production. The obtained DAT-HOF/Co-HHTP electrocatalyst delivered a superior $2e^-$ ORR performance with a H_2O_2 Faradic efficiency of 97.1%, a H_2O_2 yield of $738.9 \text{ mg h}^{-1} \text{ cm}^{-2}$ and a long-term durability for over 100 h at 1200 mA cm^{-2} . Systematical investigations revealed that the adsorption strength of key oxygen intermediates on the Co active sites was optimized by forming Co-N chemical bonds at the heterointerface. In addition, a built-in electric field was formed for accelerating proton migration from DAT-HOF to Co-HHTP, accelerating the protonation kinetics of $2e^-$ ORR. The material design of HOF@conductive MOFs heterostructure is innovative and the performance is impressive. Overall, the manuscript is recommended for publication pending minor revisions as detailed below.

Response: We thank Reviewer 3 for the positive comments.

Comment 1. The experimental conditions of flow cell test are not specified.

Response: We thank Reviewer 3 for the helpful comment. The descriptions on the experimental conditions of flow cell test have been provided in the revised manuscript as follows:

"4.2 The ORR activities evaluated in flow-type electrolytic cells

Electrochemical tests were carried out on a CHI760E electrochemical workstation connected to a CHI680D high current amplifier. A standard three-electrode three-phase flow cell system was assembled by employing gas diffusion electrode (GDE) as a working electrode, platinum electrode as a counter electrode, and Ag/AgCl (KCl, 3.5 M) as a reference electrode. The catalyst inks were prepared by dispersing 10 mg of sample into 1 mL of isopropanol containing 40 μL of Nafion solution to form a homogeneous suspension. Next, 50 μL of the homogeneous catalyst ink was dripped on GDE with an overall area of $2 \times 2 \text{ cm}^2$ and active area of $1 \times 1 \text{ cm}^2$). The catholyte and anolyte were both 0.1 M K_2SO_4 aqueous solution (pH=7). A peristaltic pump was used to circulate the electrolyte with a rotational speed of 30 rpm. The O_2 gas flow was maintained as 20 mL min^{-1} during the whole measurement. The H_2O_2 production rate was determined by the iodometry method. Typically, 100 μL of reaction solution was collected from the electrochemical system and subsequently added to the mixture of potassium hydrogen phthalate ($\text{C}_8\text{H}_5\text{KO}_4$) and potassium iodide (KI) aqueous solution with reaction for 30 min. The H_2O_2 were allowed to react with I^- to generate I_3^- ($\text{H}_2\text{O}_2 + 3\text{I}^- + 2\text{H}^+ \rightarrow \text{I}_3^- + \text{H}_2\text{O}$). The amount of I_3^- was measured by a Synergy-H1 microplate reader at its characteristic absorbance peak of 350 nm for H_2O_2 quantification."

Comment 2. The figure captions for Figure. 3e-f are missing.

Response: We are grateful for Reviewer 3's useful comment. The captions of Figure 3e and 3f have been added as below.

Figure 3. (e) UPS spectra of DAT-HOF and Co-HHTP. (f) Built-in electric field in DAT-

HOF@Co-HHTP.

Comment 3. The electron transfer numbers of the three catalysts should be provided.

Response: We thank Reviewer 3 for the professional comment. The electron transfer numbers of the catalysts have been provided in Figure S7b in the revised manuscript as below.

Figure S7b. Electron transfer numbers of different samples.

Corresponding descriptions have been added in the revised manuscript as follows:

"The electron transfer numbers were calculated by using the equation:

$$n = 4 \times \frac{I_d}{I_d + I_r/N}$$

where I_r is the ring current, I_d is the disk current and N is the current collection efficiency of the Pt ring electrode ($N = 0.256$)."

"The electron transfer numbers (n) of DAT-HOF, Co-HHTP and DAT-HOF@Co-HHTP were calculated to be 3.32, 2.89 and 2.08, respectively (Fig. S7b), manifesting the high $2e^-$ ORR selectivity of DAT-HOF@Co-HHTP."

Comment 4. In Figure 2e, the H_2O_2 yield and FE of DAT-HOF@Co-HHTP heterostructure were compared with those of reported electrocatalysts. The detailed reaction conditions of both literature and this work should be listed in a table in the supporting information.

Response: We appreciate Reviewer 3's valuable suggestion. The detailed reaction conditions of both literature and our work have been listed in the updated table as below.

Table S1. Performance comparison of DAT-HOF@Co-HHTTP with reported 2e⁻ ORR electrocatalysts.

Catalyst	Current density (A cm ⁻²)	Productivity (mg h ⁻¹ cm ⁻²)	FE (%)	Electrolyte	Ref
DAT-HOF@Co-HHTTP	1.2	738.9	97.1	0.1 M K ₂ SO ₄	This work
F-Cu-MOF	2	1082	84.9	0.6 M K ₂ SO ₄	12
L-ZnO	1	624.15	98.48	0.6 M K ₂ SO ₄	13
ZnO-v	1	621.88	98.1	0.6 M K ₂ SO ₄	14
TiO _x F _y	1	614	96.4	0.6 M K ⁺ (pH=13)	15
CoPc/CNT	0.2	112.2	88.7	0.1 M KPi	16
C-0.1M80	0.2	123.7	97.5	0.1 M KOH	17
VG array	0.1	61.3	94	0.1 M KOH	18
BP2000	0.4	211.14	83	0.01 M Na ₂ SO ₄ + 0.1 M H ₂ SO ₄	19
PTFE/CB NADEs	0.06	30.94	81.3	0.05 M Na ₂ SO ₄	20

ZnCo-ZIF-C3	0.07	78.812	85	0.1 M PBS	21
Sb-NSCF	0.05	25.364	94.7	0.1 M KOH	22
SNC	0.15	13.265	70	0.1 M HClO ₄	23
Pd/MCS-8	0.245	137.84	88.7	0.5 M K ₂ SO ₄	24
E-BPC	0.3	202.55	85.14	1 M Na ₂ SO ₄	25
Ir-Ta-Ti	0.11	64.08	91.8	0.1 M KOH	26
Mg ₃ (HITP) ₂	0.1	142.8	85	0.1 M PBS	27

Reference

12. Li, Q.; Nie, Z.; Wu, W.; Guan, H.; Xia, B.; Huang, Q.; Duan, J.; Chen, S. Water Spillover to Expedite Two-Electron Oxygen Reduction. *Adv. Mater.* **2025**, *37*, 2412039.
13. S. Ding, B. Xia, M. Li, F. Lou, C. Cheng, T. Gao, Y. Zhang, K. Yang, L. Jiang, Z. Nie, H. Guan, J. Duan, S. Chen, *Energy Environ. Sci.* **2023**, *16*, 3363.
14. S. Ding, Y. Zhang, F. Lou, M. Li, Q. Huang, K. Yang, B. Xia, C. Tang, J. Duan, M. Antonietti, S. Chen, *Mater. Today Energy* **2023**, *38*, 101430.
15. B. Xia, J. Du, M. Li, J. Duan, S. Chen, *Adv. Mater.* **2024**, *36*, 2401641.
16. Y. Lee, C. Lee, S. Back, Y. J. Sa, *Nanoscale* **2024**, *16*, 9545. 65
17. S. Jia, H. Yu, J. Na, Z. Liu, K. Lv, Z. Ren, S. Sun, Z. Shao, *ACS Appl. Mater. Interfaces* **2024**, *16*, 23099.
18. Y. Wang, R. Shi, L. Shang, L. Peng, D. Chu, Z. Han, G. I. N. Waterhouse, R. Zhang, T. Zhang, *Nano Energy* **2022**, *96*, 107046.
19. X. Zhang, X. Zhao, P. Zhu, Z. Adler, Z.-Y. Wu, Y. Liu, H. Wang, *Nat. Commun.* **2022**, *13*, 2880.
20. Q. Zhang, M. Zhou, G. Ren, Y. Li, Y. Li, X. Du, *Nat. Commun.* **2020**, *11*, 1731.
21. C. Zhang, L. Yuan, C. Liu, Z. Li, Y. Zou, X. Zhang, Y. Zhang, Z. Zhang, G. Wei, C. Yu, *J. Am. Chem. Soc.* **2023**, *145*, 7791.
22. M. Yan, Z. Wei, Z. Gong, B. Johannessen, G. Ye, G. He, J. Liu, S. Zhao, C. Cui, H.

Fei, Nat. Commun. **2023**, *14*, 368.

23. Z. Mou, Y. Mu, L. Liu, D. Cao, S. Chen, W. Yan, H. Zhou, T. S. Chan, L. Y. Chang, X. Fan, Small **2024**, *20*, 2400564.

24. L. Y. Jing, W. Y. Wang, Q. Tian, Y. Kong, X. S. Ye, H. P. Yang, Q. Hu, C. X. He, Angew. Chem., Int. Ed. **2024**, *63*, e202403023.

25. A. Byeon, J. W. Choi, H. W. Lee, W. C. Yun, W. Zhang, C.-K. Hwang, S. Y. Lee, S. S. Han, J. M. Kim, J. W. Lee, Appl. Catal., B **2023**, *329*, 122557.

26. C. Jiang, Y.-F. Fei, W. Xu, Z. Bao, Y. Shao, S. Zhang, Z.-T. Hu, J. Wang, Appl. Catal., B **2023**, *334*, 122867

27. K. Dong, J. Liang, Y. Wang, L. C. Zhang, Z. Q. Xu, S. J. Sun, Y. S. Luo, T. S. Li, Q. Liu, N. Li, B. Tang, A. A. Alshehri, Q. Li, D. W. Ma and X. P. Sun, Conductive two-dimensional magnesium metal-organic frameworks for high-efficiency O₂ electroreduction to H₂O₂, ACS Catal., **2022**, *12*, 6092-6099.

Comment 5. The DFT calculations should also involve the *OOH→*HOOH hydrogenation step to further support the facilitated hydrogenation by integrating HOF.

Response: We sincerely thank Reviewer 3 for the thoughtful comment. The free energy change of *OOH→*HOOH hydrogenation step has been added in the revised Figure 4c as below.

Figure 4. (c) Free energy diagrams for ORR over Co-HHTP and DAT-HOF@Co-HHTP heterostructure. (d) Schematic diagram of the ORR pathways over DAT-HOF@Co-HHTP heterostructure.

The following descriptions have been added in Page 15 in the revised manuscript.

"For the second hydrogenation step from *OOH to *HOOH, the energy cost for DAT-HOF@Co-HHTP (0.22 eV) is also lower than that of Co-HHTP (0.29 eV), further underscoring the crucial role of DAT-HOF in facilitating proton supply."